# The Immune Functions of Keratinocytes in Skin Wound Healing

**DOI:** 10.3390/ijms21228790

**Published:** 2020-11-20

**Authors:** Minna Piipponen, Dongqing Li, Ning Xu Landén

**Affiliations:** Center for Molecular Medicine, Ming Wai Lau Centre for Reparative Medicine, Department of Medicine Solna, Dermatology and Venereology Division, Karolinska Institute, 17176 Stockholm, Sweden; minna.piipponen@ki.se (M.P.); dongqing.li@ki.se (D.L.)

**Keywords:** keratinocyte, wound healing, immune function, cytokine, antimicrobial peptide, epigenetic regulation, microRNA, long noncoding RNA, chronic wounds, inflammation

## Abstract

As the most dominant cell type in the skin, keratinocytes play critical roles in wound repair not only as structural cells but also exerting important immune functions. This review focuses on the communications between keratinocytes and immune cells in wound healing, which are mediated by various cytokines, chemokines, and extracellular vesicles. Keratinocytes can also directly interact with T cells via antigen presentation. Moreover, keratinocytes produce antimicrobial peptides that can directly kill the invading pathogens and contribute to wound repair in many aspects. We also reviewed the epigenetic mechanisms known to regulate keratinocyte immune functions, including histone modifications, non-protein-coding RNAs (e.g., microRNAs, and long noncoding RNAs), and chromatin dynamics. Lastly, we summarized the current evidence on the dysregulated immune functions of keratinocytes in chronic nonhealing wounds. Based on their crucial immune functions in skin wound healing, we propose that keratinocytes significantly contribute to the pathogenesis of chronic wound inflammation. We hope this review will trigger an interest in investigating the immune roles of keratinocytes in chronic wound pathology, which may open up new avenues for developing innovative wound treatments.

## 1. Introduction

As the outermost barrier of our body, the skin is subjected to daily assaults from the external environment. Efficient wound repair is vital for survival, and that is achieved by the delicate orchestration of different cellular and molecular events in a temporal sequence. The wound healing process has been characterized as a series of distinct but partially overlapping phases, i.e., inflammation, proliferation, and remodeling [1]. As the most dominant cell type constituting the epidermis, keratinocytes play multiple roles essential for skin repair. They are the executors of the re-epithelialization process, whereby keratinocytes migrate, proliferate, and differentiate to restore the epidermal barrier. Transitions of these different cellular states of keratinocytes are modulated by various wound microenvironmental cues, including growth factors, cytokines, chemokines, and matrix metalloproteinases (MMPs), which have been extensively reviewed previously [2]. Moreover, keratinocytes participate in contracting wounds together with fibroblasts [3,4].

Recently, the importance of nonhematopoietic, structural cells for mammalian immunity has been highlighted [5]. With regard to this, keratinocytes have gained increasing attention for their active contribution to host immune responses in skin wound healing. Keratinocytes express various immune genes activated by the injury itself [6] or together with external pathogens. Keratinocyte-derived cytokines, chemokines, antimicrobial peptides (AMPs), and extracellular vesicles mediate the extensive interactions between keratinocytes and hematopoietic immune cells, and such crosstalk plays an essential role in driving wound healing [7]. In this review, we focus on keratinocytes’ immune functions, which have been less studied than their structural role in skin wound healing.

## 2. Crosstalk between Keratinocytes and Immune Cells during Skin Wound Healing

Keratinocytes have active crosstalk with immune cells during wound repair, which is mediated by secreted signal proteins (e.g., cytokines and chemokines, Section 2.1) and extracellular vesicles (Section 2.2) or through direct interactions (Section 2.3).

### 2.1. Keratinocyte‒Immune Cell Communication via Cytokines and Chemokines

#### 2.1.1. Pattern Recognition Receptor-Mediated Cytokine and Chemokine Expression in Keratinocytes

As the first line of defense, keratinocytes can sense specific small molecular motifs present on bacteria and other microorganisms, which are referred to as pathogen-associated molecular patterns (PAMPs) [8]. PAMPs, such as bacterial lipopolysaccharides (LPS), endotoxins, or viral-derived nucleic acids, are recognized explicitly by pattern-recognition receptors (PRRs). Keratinocytes express various PRRs, including Toll-like receptors (TLRs), C-type lectin receptors, nucleotide-binding oligomerization domain-like receptors, and retinoic-acid-inducible gene I (RIG-I)-like receptors [9]. Ligand recognition via specific PRRs leads to the subsequent activation of distinct signaling pathways and the production of inflammatory cytokines, chemokines, and host antimicrobial molecules [10]. After being released from keratinocytes, these factors serve an important function in activating skin-resident immune cells and recruiting circulating immune cells to the wound site [8].

TLR activation is a critical element in initiating and amplifying inflammation after skin injury (Table 1 and Figure 1). As a main constituent of the epidermis, keratinocytes express multiple TLRs, including TLR-1, -2, -4, -5, and -6, and the endosomal TLR-3 and -9, whose levels are significantly upregulated in acute wounds [11]. TLR2 forms a heterodimer with either TLR1 or TLR6 to recognize bacterial tri-acyl lipopeptides and di-acyl lipopeptides, respectively. TLR2 can also recognize bacterial peptidoglycan (PGN) and lipoteichoic acid. TLR4, together with CD14, recognizes LPS of Gram-negative bacteria. TLR5 is well known to specifically sense and recognize flagellin, which is a major structural protein of bacterial flagella. TLR9 is a nucleotide-sensing TLR that recognizes bacterial unmethylated 5′-cytosine-phosphate-guanosine-3′ (CpG)-containing DNA [12] and host-derived denatured DNA from apoptotic cells [11]. Depending on the adaptor usage, TLR signaling is primarily divided into MyD88-dependent and TIR-domain-containing adapter-inducing interferon-β (TRIF)-dependent pathways [13]. Both MyD88 and TRIF can induce the activation of the nuclear factor kappa-light-chain-enhancer of activated B cells (NF-kB) signaling pathway. TLR3 and TLR4 have also been shown to activate interferon regulatory factor 3 (IRF3), while TLR9 activates interferon regulatory factor 7 (IRF7) to induce type 1 interferon (IFN) expression.

Of note, TLR3 activation in the skin epithelium is essential for the normal immune response after injury [19]. TLR3 recognizes viral double-stranded RNA (dsRNA), as well as dsRNA derived from damaged cells, which results in the keratinocyte expression of cytokines, including IFN-β, interleukins 8, 18, 36γ (IL-8, IL-18, IL-36γ) and tumor necrosis factor alpha (TNF-α) [16,17,18], as well as chemokines, such as C-C motif chemokine ligands 20 and 27 (CCL20 and CCL27) [9]. Exposure to dsRNA further upregulates the expression of dsRNA-sensing receptors, such as TLR3, but also single-stranded RNA (ssRNA) sensing receptor TLR7, which is barely expressed in keratinocytes under homeostatic conditions [20,21]. It has recently been shown that dsRNA can complex with a human antimicrobial peptide (AMP), cathelicidin LL-37, to amplify TLR-mediated inflammatory signals [22,23]. Moreover, costimulation with dsRNA and cytokines, such as IL-17, IFN-γ, or IFN-α, leads to a synergistic effect on the increased expression of cytokines, AMPs, and dsRNA-sensing receptors in keratinocytes [18,24,25]. Thus, during tissue injury, dsRNA triggers a powerful positive feedback loop in keratinocytes, leading to a rapid immune response. Furthermore, the activation of TLR3 also induces keratinocytes to express genes essential for skin repair and maintenance, such as transglutaminase 1 (TGM1) and ATP binding cassette subfamily A member 12 (ABCA12) [21,26]. In addition, TLR3 activation enhances retinoic acid synthesis and the expression of hair follicle stem cell markers, such as leucine-rich-repeat containing G proteins (LGR) 5 and 6, which in turn promote hair follicle regeneration [27,28].

In addition to TLRs, RIG-I-like receptors (RLRs) are important cytosolic nucleic acid sensors in keratinocytes. When bound to dsRNA or dsDNA, they trigger an antiviral defense by activating type I IFN production [10,29]. Type I IFNs are critical regulators of host antiviral defense, as they can promote myeloid dendritic cell maturation, T-cell proliferation, and priming of CD8+ T-cells, and they stimulate B-cell differentiation into antibody-secreting plasma cells [30]. However, an uncontrolled production of type I IFNs can contribute to psoriasis pathogenesis. RIG-1 and melanoderma differentiation-associated gene 5 (MDA5) respond efficiently to dsRNA and trigger IRF3 activation and the subsequent expression of type I IFN expression in keratinocytes [21]. Other cytosolic nucleic acid sensors include cyclic guanosine monophosphate–adenosine monophosphate (cGAMP) synthase, or cGAS, and absent in melanoma 2 (AIM2), which detect dsDNA. The stimulator of interferon genes (STING) signaling plays a vital role in keratinocyte inflammatory response. Upon dsDNA binding, cGAS generates the second messenger cGAMP, which binds and activates STING and subsequent IRF3-mediated type I IFN production [29,31]. Interferon-γ-inducible protein 16 (IFI16) has been shown to co-operate with cGAS in keratinocyte dsDNA-sensing to activate the STING pathway and type I IFN production [32]. In line with this, topical treatment with cGAMP accelerates skin wound healing through enhanced expression of IFN-β in the wound sites [33].

Whereas cGAS, RIG-1, and MDA5 all contribute to type I IFN-response in keratinocytes, AIM2 detects cytosolic DNA and triggers IL-1β release in keratinocytes. A strong epidermal upregulation of AIM2 is associated with acute and chronic skin inflammatory conditions, such as psoriasis [34,35]. AIM2 expression is induced by IFN-γ in keratinocytes [34,35], as well as IFI16 and RIG-1 expression [32,36]. Therefore, by inducing the type I IFN expression, these nucleic acid sensors can upregulate themselves in an autocrine manner. IFN-α/β can activate transcription of type-I IFN-stimulated genes (ISGs) via the Janus kinase (JAK) and signal transducer and activator of transcription (STAT) pathways [31]. Additionally, the expression of pro-inflammatory cytokines TNF-α, IL-6, C-X-C motif chemokine ligand 10 (CXCL10), and CCL5 has shown to be increased by the cGAS–STING pathway [31]. Activation of the STING pathway by cGAMP leads to increased IFN-β expression in wound sites and the upregulation of CXCL10 and CCL2, which are important chemotactic factors for immune cells [33]. A recent study by Dunphy et al. shows that DNA damage can trigger a rapid innate immune response and production of IFN-β, IL-6, and CCL20 by STING but independent of cGAS and cGAMP [37]. Of note, this non-canonical activation of STING leads to the activation of NF-κB, rather than IRF3, suggesting that STING can shape the transcriptional program of immune response depending on the stimulus [37].

#### 2.1.2. Other Receptors Expressed by Keratinocytes to Receive Signals from Immune Cells

By secreting cytokines and chemokines, keratinocytes can recruit, activate, and regulate immune cells. On the other hand, keratinocytes also sense wound microenvironmental cues, including the signals from immune cells and respond with changed cellular functions [38]. The key cytokines, chemokines, and their receptors involved in keratinocyte immune function have been reviewed recently [10]. Here, we will focus on the related evidence implicated in the wound healing process.

Keratinocytes express receptors of TNF-α and IFN-γ. Stimulation with these cytokines induces the keratinocyte expression of a broad range of pro-inflammatory genes, such as CXCL5, CXCL8 (also referred to as IL-8), and intercellular adhesion molecule 1 (ICAM-1, or CD54) [9,39]. Additionally, IFN-γ upregulates major histocompatibility complex (MHC) class II expression in keratinocytes [9,40], which are important for antigen presentation to T cells and will be further discussed in Section 2.3.

IL-1α and IL-1β are important pro-inflammatory cytokines produced by many cell types, including keratinocytes, which constitutively express IL-1α, and to a less extent IL-1β. IL-1 expression is quickly increased upon skin injury, and IL-1α binding to the IL-1 type I receptor induces keratinocytes to express CCL20 to attract immune cells [41,42]. IL-1β induces skin antimicrobial defense, and, together with IL-6 and IL-23, it is crucial for IL-17-producing T cells [42].

IL-6 has various effects on different cells to promote wound healing. Upon skin injury, the expression of IL-6 and its receptor is increased in keratinocytes [41]. The IL-6 receptor consists of the IL-6 receptor subunit (IL-6R) and IL-6 signal transducer glycoprotein 130 (gp130). Its activation by IL-6 induces keratinocyte proliferation via STAT3 signaling [42]. In addition, keratinocytes express a soluble IL-6 receptor (sIL-6R), which is responsive to IL-6 to promote skin barrier repair [43].

IL-17 and IL-22 are mainly produced by immune cells, and they have a pivotal role in immunoregulation and host defense. IL-17 signaling is mediated through heterodimeric IL-17 receptor A/C (IL-17RA/C) and A/D (IL-17RA/D) to induce IL-23 and AMP expression in keratinocytes, and upon skin injury, keratinocytes upregulate IL-17RA [42,44]. IL-22 signaling is mediated through heterodimeric IL-22 receptor (IL-22R) and IL-10 receptor β2 (IL-10Rβ2) in keratinocytes to activate STAT3 signaling [42].

The healthy human epidermis has been shown to express chemokine receptors C-C chemokine receptor (CCR)1, CCR3, CCR4, CCR6, CCR10, C-X-C Motif Chemokine Receptor (CXCR)1, CXCR2, CXCR3, and CXCR4 [45]. For five of them, i.e., CCR1, CCR10, CXCR1, CXCR2, and CXCR3, binding with their corresponding monospecific ligands, i.e., CCL14, CCL27, CXCL8, CXCL1, and CXCL10, respectively, can stimulate keratinocyte migration and proliferation as well as wound re-epithelialization [45]. These chemokines are secreted by both immune cells and keratinocytes after wounding [45].

#### 2.1.3. Keratinocyte–Immune Cell Crosstalk in Skin Wound Healing

For wound healing, it is critical to coordinate the activities of different cell players accurately. The active crosstalk between keratinocytes and immune cells that has been detected through various stages of wound repair is one such example (Figure 2).

##### Neutrophils

Upon injury, keratinocytes produce pro-inflammatory cytokines (e.g., TNF-α and IL-1) and chemokines (e.g., CXCL1, -5, -8), which play an important role in the recruitment and activation of neutrophils and macrophages at the wound site [46]. Neutrophils are the first immune cells to arrive at affected tissue, where they remain for 2‒5 days before undergoing apoptosis if no infection occurs [47]. Neutrophils kill invading pathogens by producing reactive oxygen species (ROS), AMPs, and proteases, and by phagocyting them with web-like neutrophil extracellular traps (NET) [46,47]. Neutrophils produce cytokines (e.g., TNF-α, IL-1β, and IL-6) and chemokines (e.g., CXCL2, CXCL8, and monocyte chemoattractant protein 1, MCP-1, also referred to as CCL2) to attract more immune cells, thus amplifying the inflammatory response [48]. In part, neutrophil-secreted TNF-α, IL-1β, and IL-6 also promote keratinocyte immune responses and proliferation [10,42,43,49]. After completing their mission, apoptotic neutrophils express phosphatidylserine (PS), Annexin I, and calreticulin on their cell surface, which serve as signals to initiate neutrophil engulfment by macrophages, diminishing the inflammatory phase [50]. 

##### Macrophages

Monocytes infiltrate to the wound site after neutrophils and differentiate into macrophages with a pro-inflammatory phenotype (M1 macrophages), removing cellular debris, microbes, and apoptotic neutrophils [46]. Similar to neutrophils, M1 macrophages produce ROS and pro-inflammatory factors, such as IL-1, IL-6, and TNF-α, as well as matrix metalloproteinases, to break down the extracellular matrix [51]. M1 macrophage-produced IL-6 and TNF-α also stimulate fibroblasts to secrete keratinocyte mitogens, such as keratinocyte growth factor (KGF)-1, -2, and IL-6 [49]. The conversion from the pro-inflammatory (M1) to anti-inflammatory (M2) phenotype of macrophages, which is promoted by their engulfment of apoptotic material, is essential to impede inflammation and to initiate the proliferative phase of wound healing [52]. M2 macrophages produce several factors to collectively promote tissue repair, including transforming growth factor beta (TGF-β, fibroblast activation and extracellular matrix synthesis), MMPs (tissue remodeling), vascular endothelial growth factor (VEGF, neovascularization), and platelet-derived growth factor (PDGF, cell proliferation) [46,48,51].

##### Langerhans Cells (LC)

LCs constitute a dendritic cell population of antigen-presenting cells that play a central role in maintaining immune homeostasis in the skin [53]. Keratinocyte-derived IL-34, TGF-β, and receptor activator of nuclear factor kappa-Β ligand (RANKL) are important for LC growth and differentiation [54]. Upon tissue injury, keratinocytes secrete pro-inflammatory cytokines, such as IL-1β, TNF-α, and granulocyte-macrophage colony-stimulating factor (GM-CSF) [55]. These factors enhance LC migration from the skin to the draining lymph nodes, where they can prime naïve T cells, thus bridging the innate and adaptive immune responses [53,56]. Activated T cells are recruited to the wound site, and soon after, the epidermis is repopulated by short-lived monocyte-derived LCs, which are later replaced by a more permanent population of LCs [57,58]. GM-CSF is also a potent activator of keratinocyte proliferation and has been shown to accelerate wound healing. GM-CSF overexpression in mice leads to increased re-epithelization and wound closure, whereas GM-CSF depletion impairs wound healing significantly [59,60]. Moreover, higher cellularity and earlier granulation tissue formation are seen in GM-SCF overexpressing mice, suggesting a more rapid filtration of immune cells to the wound site [60]. Clinically, the topical use of recombined human GM-CSF has been shown to treat burn wounds and chronic leg ulcers [61].

##### CD8^+^ Tissue-Resident Memory T (TRM) Cells

TRM cells are a noncirculating population of T cells that reside in the tissue, providing immunosurveillance and a rapid response against pathogen invasion [62]. CD8^+^ T_RM_ cells are not only present in the dermis but are also located in tight contact with keratinocytes, particularly within the hair follicle epithelium. Keratinocyte-secreted factors, such as interleukins and chemokines, are necessary for TRM cell homeostasis and recruitment upon infection [63,64,65]. On the other hand, CD8^+^ TRM cells secrete IFN-γ to modulate the chemokine synthesis of keratinocytes—for instance, the induction of CXCL9 and CXCL10 [66]. CD8^+^ TRM cells signal through keratinocytes to provide active immune protection. For example, normal skin flora *Staphylococcus epidermidis* induces CD8^+^ T cell accumulation and the production of IL-17 and IFN-γ in the mouse epidermis. These CD8^+^ T cells preferentially localize to the basal epidermis or close to the epithelial layer and express CD103 and CD69, which are cell surface markers for skin TRM cells [65,67]. IL-17 and IFN-γ activate keratinocytes to release S100A8 and S100A9 [67], which have antimicrobial activity and can also function as chemoattractants for neutrophils. Moreover, *S. epidermidis*-activated CD8^+^ T cells express several keratinocyte mitogens, such as amphiregulin (AREG), fibroblast growth factor 2 (FGF2), and TGFB1 [68]. In line with this, wound healing is accelerated in mice previously associated with *S. epidermidis* [68].

##### Dendritic Epidermal T Cells (DETCs)

DETCs are a unique subset of γδ T cells in mice that reside in tight contact with keratinocytes and react rapidly to epidermal injury [69]. In wound keratinocytes, the expression of specific ligands recognized by DETCs, such as H60c (ligand for cytotoxic lymphocyte activating receptor NKG2D), plexin B2, and coxsackievirus and adenovirus receptor (CAR), are upregulated [70,71], which promote the activation and infiltration of immune cells into the epidermis [72]. The depletion of NKG2D or inhibition of H60c binding to NKG2D has been shown to inhibit wound healing in mice [73]. Additionally, DETCs secrete keratinocyte growth factors, e.g., KGF, KGF2, and insulin growth factor 1 (IGF1), to promote keratinocyte proliferation and wound re-epithelization. DETC-produced IL-17 and KGF also activate keratinocytes to produce AMPs and hyaluronan [74,75], enhancing the immune response. In the absence of DETCs, wound healing is severely impaired in the mouse skin, as a lack of DECT-derived keratinocyte growth factors results in keratinocyte apoptosis [76]. The human epidermis contains both γδ and αβ T cells, but they are not directly analogous to mouse DECTs [69]. However, human γδ and αβ T cells exhibit a similar function to DETCs by actively producing IGF-1 to promote wound healing [77]. In line with this, T cells isolated from human chronic wounds are functionally impaired to produce IGF-1 [77].

### 2.2. Keratinocyte Crosstalk with Immune Cells via Extracellular Vesicles

Cells can extend their communication with other cells by delivering bioactive cargo, such as proteins, lipids, DNA, and RNA, packed in specific membrane-enclosed vesicles called extracellular vesicles (EVs) [78]. EVs are categorized into three subclasses based on their biogenesis: apoptotic bodies (1‒5 µm), microvesicles (100‒1000 nm), and exosomes (<150 nm). They can be distributed locally or to distant sites through the bloodstream. A growing number of studies have shown that EVs can regulate many biological processes by releasing their content to the recipient cell or acting on the cell surface by ligand‒receptor interaction without cargo delivery [78].

Keratinocyte EV secretion is induced by various stimuli, such as hypoxia, irradiation, and starvation. Increasing evidence depicts that keratinocyte-derived EVs play a role in the wound healing process [79,80]. Keratinocyte-derived exosomes can affect the dendritic cell phenotype and cytokine production [81]. Cytokine-treated keratinocytes secrete exosomes that promote neutrophil pro-inflammatory factor production, including IL-6, IL-8, and TNF-α [82].

A study by Zhou et al. shows that exosome secretion is significantly induced after injury [83]. Wound macrophages selectively engulf exosomes derived from the wound-edge keratinocytes due to the specific surface N-glycan on exosomes [83]. These exosomes are highly abundant with small RNAs (<100 bp), suggesting they carry miRNAs. Consistent with this, the specific inhibition of miRNA packaging into exosomes resulted in the accumulation and persistence of wound macrophages in the granulation tissue even after wound closure, indicating close crosstalk between keratinocytes and macrophages through exosome-delivered miRNAs [83]. A closer examination of miRNA composition was not performed in this study, but a recent study by Than et al. suggests that miRNAs undergo selective sorting into distinct types of EVs in keratinocytes under homeostatic conditions [84].

The administration of exosomes derived from mesenchymal stem cells (MSCs) or adipose tissue stem cells has been shown to improve wound healing by promoting fibroblast proliferation and migration, collagen synthesis, and angiogenesis [85,86,87,88,89,90]. Additionally, a phenotypic shift from pro-inflammatory M1 macrophages to anti-inflammatory M2 macrophages by M2 macrophage-derived exosomes has been shown to accelerate wound healing [27].

There is a growing interest in utilizing EVs as a therapeutic tool to treat nonhealing wounds. Genetic engineering or the pre-treatment of EV donor cells or direct drug loading into exosomes are possible strategies in creating EV-based therapies. A phase 1/2A clinical trial is ongoing to assess the safety and efficacy of MSC-derived exosomes to treat dystrophic epidermolysis bullosa, which is a genetic and severe skin blistering disorder (NCT04173650). Moreover, several studies have shown the pro-healing effects of exosomes in animal models with impaired healing capacity [86,88,91,92,93,94].

### 2.3. Keratinocyte Interaction with T Cells via Antigen Presentation

In addition to producing cytokines and EVs, keratinocytes can also function as atypical antigen-presenting cells (APC) to interact with T lymphocytes in an antigen-specific manner. MHC-II expression has been detected in keratinocytes from a wide variety of skin diseases, such as lupus erythematosus, vitiligo, lichen planus, cutaneous T-cell lymphoma, various infectious dermatoses, allergic contact dermatitis, granulomatous dermatoses, lichen sclerosis, erythema nodosum [95], and psoriasis [96]. It is known that keratinocytes can synthesize HLA-DR under the stimulation of IFN-γ [97]. In the presence of alloantigen or nominal antigen, the interaction between MHC-II+ keratinocytes and T cells results in T cell anergy or tolerance [98,99], while bacterial-derived superantigens presented by keratinocytes activate T cells to produce mainly IL-4, IL-5, and IL-10, but not IFN-γ [100]. A small fraction of keratinocytes expressing MHC-II was also found in humans [101] and mouse epidermis under homeostasis [102]. In mouse skin, MHC-II^+^ keratinocytes were found to control the homeostatic type 1 response to the microbiota [102]. In addition, epithelial MHC-II expression and antigen-presenting function have also been reported in the gastrointestinal and respiratory tracts [103]. Interestingly, a recent report shows that mouse intestinal stem cells’ (ISC) T cell interaction via MHC-II not only allows ISC to regulate immune responses but also enables the immune system to modulate ISC renewal and differentiation [104].

## 3. The Interplay between Keratinocytes and Microorganisms in Skin Wound Healing

### 3.1. The Role of Microbiota in Wound Healing

Skin microbiota plays a vital role in many aspects of human health [105]. Its impact is especially prominent during skin injury when the microbiota is translocated from the skin surface to the wounded skin’s dermis, which triggers the inflammatory response. The overall impact of skin microbiota on wound repair remains controversial. It has been shown that wound healing is accelerated in the absence of commensal bacteria, as the inflammation is reduced, which, in turn, promotes angiogenesis in the wound bed [106]. Conversely, several studies support a beneficial role for skin commensal bacteria in wound healing [68,82,107]. For instance, lipoteichoic acid (LTA) produced by *S. epidermidis* exhibits an anti-inflammatory effect on keratinocytes and restricts excessive inflammatory response upon tissue injury [19]. *S. epidermidis* also secrets lipopeptide 78 (LP78), which inhibits TLR3-mediated skin inflammation and promotes wound healing. In contrast, another lipopeptide (LP01) from these bacteria activates TLR2/CD36-p38 mitogen-activated protein kinase (MAPK) to enhance antimicrobial defense against pathogenic infections [108,109]. A deeper understanding of the underlying mechanisms is vital for developing new therapeutic strategies for chronic nonhealing wounds, particularly for those with biofilm infection.

### 3.2. AMPs and Wound Healing

In addition to cytokines and chemokines, the activated keratinocytes also produce large amounts of AMPs, such as β-defensins, cathelicidins, S100 proteins (e.g., psoriasin and calprotectin), and RNase 7, which are positively charged amphiphilic molecules and can kill various microbial pathogens directly [110]. For instance, *Escherichia coli* induces psoriasin (S100A7) secretion by keratinocytes, while *S. aureus* stimulates keratinocytes to produce β-defensins, and these AMPs protect the skin from infections [111,112,113]. Interestingly, AMPs are also produced by the skin commensal microbiome, e.g., *S. epidermidis* produces AMPs to selectively kill and prevent pathogenic bacteria *S. aureus* colonization [114]. Moreover, the skin commensal microbiome can regulate the adaptive immune response by modulating host AMP production [115].

AMPs are expressed through wound healing stages and contribute to wound repair in many aspects [116,117]. As a normal response to skin injury, keratinocytes produce higher amounts of cathelicidin antimicrobial peptide LL-37, whereas, in chronic wounds, there is a lack of LL-37 [118,119]. Consistently, LL-37 inhibition impairs wound healing [118], while its adenoviral delivery to mice promotes wound re-epithelialization and granulation tissue formation [120]. In addition to its antimicrobial activity, LL-37 can function in chemotaxis and enhance TLR3-signaling in complex with dsRNA analog polyinosinic:polycytidylic acid (poly(I:C)) [110]. Keratinocytes constitutively produce β-defensin 1 (hBD1), whereas hBD2, -3, and -4 are induced in response to injury to promote keratinocyte migration and proliferation but also to stimulate keratinocyte pro-inflammatory cytokine and chemokine production (e.g., IL-6, IL-10, CCL-5) [119]. Perforin-2 (or macrophage-expressed gene 1, MPEG1) is an antibacterial effector protein constitutively expressed in the epidermis and upregulated after wounding [56]. Increased clearance of intracellular *S. aureus* is observed in perforin-2 overexpressing keratinocytes [56], suggesting that keratinocytes can utilize perforin-2 in bacterial killing during wound healing. While many AMPs actively promote re-epithelization and granulation tissue formation, some of them also have a role in collagen synthesis at the remodeling phase, including AH90, SR0379, and epinecidin-1 [110].

## 4. Epigenetic Regulation of Keratinocyte Immune Functions in Wound Healing

To carry out the immune functions discussed above, keratinocytes need to regulate their gene expression accurately and efficiently in response to various external stimuli. In this regard, epigenetic mechanisms have gained increasing attention in understanding gene expression regulation during wound healing [121]. The epigenome comprises regulatory molecules and chemical marks modulating genomic activity without changing the DNA sequence, which is an important interaction between the genome and the environment. Major epigenetic mechanisms include histone modifications, DNA methylation, non-protein-coding RNAs, and three-dimensional chromatin organization [122,123]. In this review, we focus on the epigenetic mechanisms that regulate keratinocyte immune functions (Figure 3).

### 4.1. Histone Modifications

Histones are the critical proteins involved in DNA assembly into chromatin, and their post-transcriptional modifications have a fundamental impact on chromatin structure and gene expression [122]. Several types of histone modifications exist, including acetylation, methylation, phosphorylation, ubiquitylation, and sumoylation. The histone modification process is highly dynamic and can directly alter the histone‒DNA interaction and chromatin compactness as well as regulate the binding of chromatin-modifying protein complexes to histone tails. There is complex crosstalk between different histone modifications to control the overall chromatin organization [122].

The Jumonji domain containing-3 (JMJD3) is an epigenetic regulator that specifically catalyzes methyl groups’ removal from lysine residues on histone tails, releasing a repressive epigenetic mark (H3K27me3) and activating gene expression. JMJD3 has been found to be upregulated in wound-edge keratinocytes. In cooperation with NF-κB, JMJD3 activates the expression of several inflammatory genes, such as IL-1β, IL-6, CCL20, and TNF-α, by demethylating their gene promoters in scratch-wounded keratinocytes in culture [124]. Moreover, upon skin injury, Notch1, a key transcription factor in skin development and homeostasis, is activated in keratinocytes and induces TNF-α and chemokine expression [125]. Interestingly, Notch1 has been identified as a direct target of JMJD3 and NF-κB in wound-edge keratinocytes [126], suggesting a trigger function for JMJD3 in the keratinocyte immune response. By decreasing the repressive histone methylation mark in the gene promoter, JMJD3 also promotes the pro-inflammatory (M1) phenotype and IL-12 production in macrophages in diabetic wounds [127]. 

### 4.2. Trained Immunity of Epithelial Stem Cells

“Trained immunity” (TI) or “innate immune memory” is a novel concept describing the capacity of an organism to develop increased responsiveness to secondary stimuli independent of adaptive immunity [124]. TI was first discovered in innate immune cells, e.g., monocytes, macrophages, and natural killer cells; however, these cells in circulation have a lifespan shorter than the duration of TI. Indeed, TI has also been found in cells with long lifespans, such as stem cells and fibroblasts [128]. Interestingly, Naik et al. showed that epithelial stem cells (EpSCs) could harbor a long-lasting memory of previous inflammatory stimuli, i.e., a topical imiquimod treatment, enabling the skin to respond to subsequent assaults swiftly and accelerates wound repair [129]. Chromatin dynamics have been shown to mediate the TI of EpSCs. After the initial stimulus, EpSCs maintain chromosomal accessibility of several critical inflammatory response genes, such as *AIM2*, allowing a rapid transcription of *AIM2* and its downstream effector genes upon secondary stimulus, i.e., skin injury [129].

### 4.3. Non-Protein-Coding RNAs

In the human genome, the majority of the transcriptional output is constituted by the RNAs that lack protein-coding capacity [130,131]. Intensive research in the recent decade has revealed that these noncoding RNAs (ncRNAs) constitute an important layer of epigenetic regulation and can function as important regulators of cellular physiology and pathology. MicroRNAs (miRNAs) and long noncoding RNAs (lncRNAs) are two of the most studied ncRNA classes. MiRNAs are ≈22-nucleotide ncRNAs. They incorporate into the RNA-induced silencing complex (RISC) and bind to the 3′ untranslated region (UTR) of the target mRNA, which results in translational repression or degradation of target mRNA [132]. Compared to miRNAs, lncRNAs are less conserved, larger in size, and exhibit more versatile gene regulatory functions, such as chromatin remodeling, transcriptional, and post-transcriptional gene regulation, protein translation, and transport [133]. Here, we will focus on the miRNAs and lncRNAs that have been shown to modulate keratinocyte immune functions.

#### 4.3.1. microRNAs

MiRNAs have been shown to regulate many different cellular processes spanning the entire wound healing process [134,135,136]. Among these wound healing-related miRNAs, several have been shown to regulate the keratinocyte production of inflammatory cytokines and chemokines (Table 2). For example, miR-132, whose expression is upregulated in human wound-edge keratinocytes, can repress chemokine (e.g., IL-8, CXCL5, CXCL1, and CCL20) and cytokine (e.g., IL-1A, IL-1B, and TNF) production but promotes the proliferation and migration of keratinocytes, thus facilitating the transition from the inflammatory to the proliferative phase of wound repair [127]. A novel human miRNA candidate, Seq-915_x4024, has been found to be highly expressed in fetal keratinocytes during early gestation. It inhibits the keratinocyte production of TNF-α, IL-6, IL-8, CXCL1, and CXCL5 while enhancing the proliferation of keratinocytes and their ability to promote fibroblast migration and growth, thus reducing scar formation and contributing to skin regeneration [137]. Similarly, miR-149 can also contribute to the scarless healing of fetal skin partially by downregulating keratinocyte inflammatory cytokine expression [138].

Moreover, the expression of some miRNAs is controlled by inflammation signals activated upon skin injury, and in turn, these miRNAs can regulate keratinocyte proliferation and migration (Table 2). The stimulation of keratinocytes by LPS triggers an inflammatory response and the expression of miR-17 while silencing miR-17 decreases cell viability and stimulates keratinocyte cytokine expression [139]. The levels of miR-31, miR-23b, and miR-223 are increased in keratinocytes after wounding [140,142,143]. MiR-223 suppresses NF-κB activation specifically in basal epithelial cells and dampens excessive neutrophil recruitment to the site of injury in a zebrafish wound model [143]. MiR-31 promotes keratinocyte proliferation and migration by activating MAPK signaling, and epidermis-specific depletion of miR-31 delays wound healing in vivo [142]. In addition, miR-23b promotes wound healing by increasing keratinocyte proliferation and suppressing cytokine production [140]. The miRNAs that regulate keratinocyte immune functions and play a pathological role in chronic wounds are summarized in Section 5.4.

#### 4.3.2. lncRNAs

Following an inflammatory stimulus, several lncRNAs show altered expression, and a growing number of these lncRNAs have been functionally characterized in mediating immune responses [151]. However, very little is known about the role of lncRNAs in regulating keratinocyte immune functions. PRINS (psoriasis-associated non-protein-coding RNA induced by stress) is an immune-responsive lncRNA in keratinocytes that was initially found in psoriasis [152]. Its expression is decreased in keratinocytes upon inflammasome activation, and it can reduce IL-6 expression by binding to IL-6 mRNA [153]. Nevertheless, its role in wound healing remains unknown. Additionally, we recently identified an lncRNA, WAKMAR2 (wound and keratinocyte migration associated lncRNA 2), which can repress the production of inflammatory chemokines (e.g., CXCL8, CCL20, and CXCL5) by keratinocytes while enhancing cell migration [154].

## 5. Impaired Immune Functions of Keratinocytes in Chronic Nonhealing Wounds

Chronic wounds are defined as cutaneous wounds that fail to heal within three months [155]. This disease not only has a profound psychological and physical impact on patients but also presents a substantial economic burden to society [156]. According to the underlying disorders, the majority of chronic wounds fall into three categories: diabetic foot ulcers (DFU), venous/arterial ulcers (VU/AU), and pressure ulcers (PU). Chronic wounds do not follow the well-defined wound healing cascades. For example, they often fail to transit from the inflammatory phase to the proliferative phase [1]. Chronic wounds are commonly featured as persistent inflammation, bacterial colonization, impaired reepithelization, and angiogenesis, as well as excessive ROS [1,46,157]. Keratinocytes at nonhealing edges of chronic wounds are hyper-proliferative but nonmigratory, and the related pathological mechanisms are being extensively studied. In this review, we focus on the dysregulated immune functions of keratinocytes in chronic wounds.

### 5.1. Cytokines

As the primary cell type in the epidermis, keratinocytes produce large amounts of cytokines to initiate and regulate inflammation, including IL-1, -6, -7, -8, -10, -12, -15, -18, and -20, and TNF-α [158]. Although elevated IL-8 [159], IL-1 α [160], TNF-α [161], IL6 [19,162], and CCL2 [163] levels have been observed in human chronic wounds, these cytokines are not specifically produced by keratinocytes. Indeed, few studies have provided clear evidence of the contribution of keratinocytes-derived cytokines to wound healing or chronic wound pathology. IL8 is a vital neutrophil chemotactic factor and induces neutrophils to migrate to the site of IL8 production, while CCL2 exhibits a chemotactic activity for macrophages. The elevated number of macrophages and neutrophils in chronic wounds may be partially due to the dysregulated cytokine expression in keratinocytes at the nonhealing wound edge [144].

### 5.2. TLRs

The expression of TLR1, TLR2, TLR4, TLR6, TLR9, and MyD88 has been found to be significantly upregulated in the wounds of diabetic patients compared with nondiabetic wounds [164,165]. We also show that the TLR3 protein level is elevated in the wound-edge epidermis of human diabetic foot ulcers, venous ulcers, and pressure ulcers, contributing to the increased neutrophil and macrophage infiltration in these chronic wounds [144]. A single-nucleotide polymorphism (SNP) study revealed that TLR4 and TLR9 SNPs and their haplotypes may increase the risk of impaired wound healing in type 2 diabetic patients [166,167]. In line with this, in a streptozotocin (STZ)-induced diabetes mouse model, knockout of TLR2 improved wound healing and reduced oxidative stress, MyD88 signaling, NF-κB activation, and cytokine secretion, suggesting that sustained TLR2 expression and activation may be detrimental to diabetic wounds [168]. Similar results were obtained from STZ-induced diabetic mice with TLR3 or TLR4 knockout, which exhibited significantly accelerated wound healing accompanied by reduced inflammatory cytokine expression [27,169,170]. Of note, these mice had TLR knocked out in all the tissues, not specifically in the epidermis; therefore, the relative contribution of keratinocytes to the observed phenotype warrants further study.

### 5.3. AMPs

The expression of AMPs is dysregulated in chronic wounds. For example, in the wound-edge epidermis of venous ulcers, psoriasin (S100A7) and hBD-2 expression was strongly induced, whereas no RNase 7 or LL-37 was detected there in comparison with the intact skin [171]. S100A8 and S100A9 were also found to be upregulated in the epidermis of human chronic wounds [172]. A recent study showed that an antimicrobial protein regenerating islet-derived 3 alpha (REG3A) decreased in the wound-edge keratinocytes of DFU patients. REG3A induces Src homology region 2 domain-containing phosphatase-1 (SHP-1) expression, which, in turn, inhibits TLR3-mediated skin inflammation under diabetic milieu [169]. Due to the beneficial role of AMPs in wound healing, they have been considered therapeutic agents for the treatment of chronic wounds. A randomized, placebo-controlled clinical trial showed that topical treatment with LL-37 enhanced the healing of hard-to-heal venous leg ulcers [173]. Additionally, an injection of regenerating islet-derived protein 3 gamma (Reg3g) has been shown to accelerate wound healing in diabetic mice.

Chronic wounds are frequently colonized by bacteria living in a biofilm, which may also contain other microorganisms, such as fungi or viruses. These microbial cells have increased resistance to antimicrobial treatment, as they are physically protected from the host immune attack by a self-produced structured polymer matrix, which contributes to a further influx of immune cells and prolonged inflammation [174]. Keratinocyte-secreted AMPs can help to prevent biofilm formation; however, deregulated AMP production has been noted in venous ulcer and burn wounds [118,171,175], making them more susceptible to defective microbial defense. Furthermore, the wound immune cells generate high levels of ROS and proteases, which lead to tissue damage and destruction of the molecules essential for wound healing, such as keratinocyte-secreted AMPs [174]. Biofilm bacteria can also suppress host AMP expression in the wound [176], and the soluble factors produced by the bacteria have been shown to be toxic to keratinocytes [177].

### 5.4. MicroRNAs

Multiple miRNAs are dysregulated in human chronic wounds and contribute to delayed wound healing through regulating keratinocyte immune functions, e.g., miR-132, miR-146, miR-34a/c, miR-19a/b, miR-20a, miR-203, miR-21, and miR-130 [127,144,145,146,148,150] (Table 2). The miR-132 expression is significantly reduced in human diabetic ulcers compared with normal acute wounds [178]. The local injection of miR-132 mimics accelerated wound closure in diabetic mice, which was accompanied by a decreased inflammatory response and increased proliferation of wound-edge keratinocytes [178]. MiR-146 has been identified as an important regulator of keratinocyte innate immunity [179]. Xu et al. showed that miR-146a expression was downregulated in diabetic mouse wounds and was negatively correlated with the increased expression of pro-inflammatory genes [145]. MiR-203 is enriched explicitly in keratinocytes, and knockdown of miR-203 in diabetic rats promoted wound healing by regulating keratinocyte proliferation [180], migration, and the epithelial to mesenchymal transition (EMT) process through its target gene IL-8 [148]. The miR-34 family is well known for its tumor-suppressive functions; however, the clinical trial of miR-34 replacement therapy in cancer patients has been halted due to severe immune-related events [146]. We found that miR-34a and miR-34c were significantly increased in the wound-edge keratinocytes of venous ulcer patients [146]. These two miRNAs promote keratinocyte inflammatory chemokine and cytokine production by targeting leucine rich repeat containing G protein-coupled receptor 4 (LGR4) [146]. We also showed that the levels of all miR-17∼92 cluster members, i.e., miR-17, miR-18a, miR-19a, miR-19b, and miR-20a, decreased in several types of human chronic ulcers, including pressure ulcers, venous ulcers, and diabetic foot ulcers, in comparison with normal acute wounds [144]. Moreover, miR-19a/b and miR-20a were found to reduce keratinocyte production of inflammatory chemokines and cytokines by regulating the TLR3-NF-κB signaling pathway by targeting SHC binding and spindle associated 1 (SHCBP1) and semaphorin 7A (SEMA7A) [144].

## 6. Conclusions and Prospects

As the most dominant cell type in our skin, keratinocytes play critical roles in skin wound repair, not only as structural cells, but also exerting important immune functions. Upon skin injury, keratinocytes are at the frontline of defense in innate immunity. At the early phase of wound healing, under the stimulation of injury and invading microorganisms, keratinocytes release large amounts of cytokines, chemokines, AMPs, and extracellular vesicles, which recruit and activate immune cells, as well as kill pathogens directly. Keratinocytes not only initiate and intensify the immune response, they also have the mechanisms to restrict and reduce inflammation, which is important for helping the wound healing process to enter the proliferative phase. Keratinocytes sense different wound microenvironmental cues and change their states (e.g., migration, proliferation, differentiation) and immune functions accordingly. The keratinocyte‒immune cell crosstalk loops are often dysregulated in chronic wounds, which are “stuck” in a chronic inflammation status: on the one hand, the inflammation is persistent, damages normal tissue, and hinders wound healing; on the other hand, it is ineffective to combat infection and to activate the signals needed for the following proliferative phases. Therefore, how to disrupt this pathological inflammatory circuit and reactivate the healing program has become the prime concern when treating chronic wounds.

Although impaired immune functions of keratinocytes have been reported in chronic wounds, their relative contribution to chronic wound pathology remains elusive. Based on their crucial immune functions in the skin, we postulate that keratinocytes are unlikely to be a sole victim in this case. They may make a significant contribution to the pathogenesis of chronic wound inflammation. However, there is not enough evidence to endorse this idea, as most previous studies on chronic wound inflammation focused on immune cells such as neutrophils and macrophages. Another intriguing question is whether the correction of keratinocytes’ immune dysfunction may help recover the immune balance in the wound microenvironment and restart the suspended healing program. A recent study highlighted the vital role of structural cells, e.g., epithelium, endothelium, and fibroblasts, in regulating the organ-specific immune response [5]. As an emerging research area, structural immunity, which aims to dissect the immune functions in the nonhematopoietic, structural cells of the body, has attracted increased attention. We hope this review will trigger broad interest in studying keratinocytes’ immune functions in wound repair and chronic wound pathology, which will open up new avenues for developing innovative wound treatments.

## Figures and Tables

**Figure 1 ijms-21-08790-f001:**
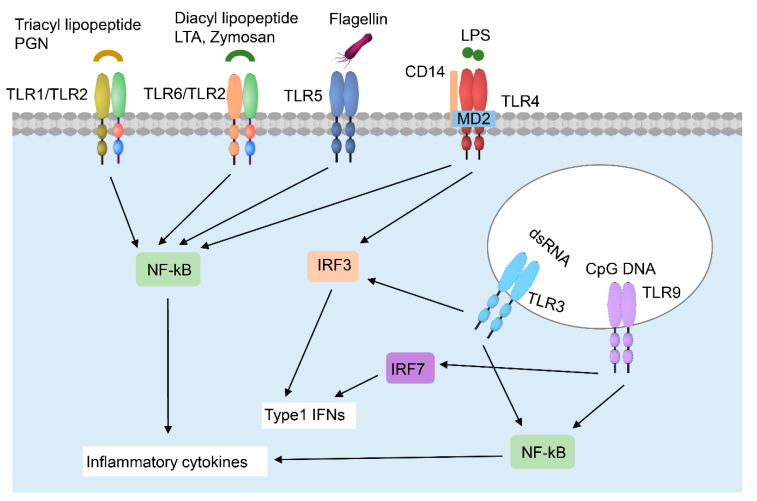
Toll-like receptors (TLR)-mediated signaling pathway in keratinocytes. TLR1, 2, 4, 5, and 6 are present on the cell membrane, while TLR3 and 9 reside in keratinocytes’ intracellular compartments. TLR2 recognizes a broad range of pathogen-associated molecular patterns, including triacyl lipopeptide (in combination with TLR1), diacyl lipopeptide (in combination with TLR6), peptidoglycan (PGN), lipoteichoic acid (LTA), and Zymosan. TLR5 recognizes flagellin, while TLR4 recognizes lipopolysaccharides (LPS), which is aided by two accessory proteins cluster of differentiation (CD14) and myeloid differentiation factor 2 (MD2). TLR3 and TLR9 reside on the endosomal membrane and recognize double-stranded RNA (dsRNA) and unmethylated CpG DNA, respectively. Upon stimulation, these TLRs activate the nuclear factor kappa-light-chain-enhancer of activated B cells (NF-kB) signaling pathway to induce the expression of inflammatory cytokines. TLR3 and TLR4 can also activate interferon regulatory factor 3 (IRF3), while TLR9 activates interferon regulatory factor 7 (IRF7) to induce the expression of type 1 interferons (IFNs).

**Figure 2 ijms-21-08790-f002:**
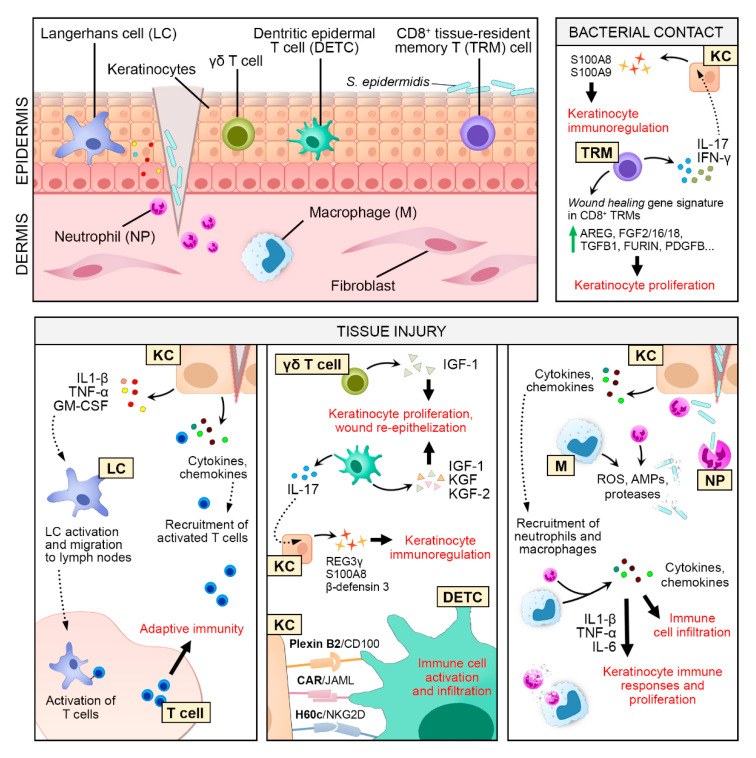
Keratinocyte‒immune cell crosstalk in skin wound healing. Keratinocyte (KC)-secreted pro-inflammatory cytokines and chemokines contribute to the recruitment of neutrophils (NP) and macrophages (M) to the site of injury. NP and M remove cellular debris and destroy invading pathogens by phagocytosis and the release of reactive oxygen species (ROS), proteases, and antimicrobial proteins (AMPs). They further amplify the inflammatory response by secreting various pro-inflammatory cytokines and chemokines to recruit more immune cells. Some of these secreted factors, such as interleukin 1 beta (IL-1β), tumor necrosis factor (TNF-α), and interleukin 6 (IL-6) stimulate keratinocyte proliferation and immune responses. After tissue injury, Langerhans cells (LC) react to KC-derived factors and migrate to the draining lymph node to activate T cells, triggering an adaptive immune response. KC-derived cytokines and chemokines serve as a chemoattractant to recruit activated T cells to the wound. Epidermal γδ T cells (human) and dendritic epidermal T cells (DETCs, mouse) respond rapidly to tissue injury and produce growth factors to activate KC proliferation and wound re-epithelization. Vice versa, ligands expressed on the KC cell membrane, i.e., Plexin B2, coxsackievirus and adenovirus receptor (CAR), and H60c are detected by DETCs, triggering immune cell activation and infiltration. DETC-produced interleukin 17 (IL-17) also activates KC to produce antimicrobial peptides (AMPs). Even in the absence of tissue injury, bacterial contact with skin commensal *Staphylococcus epidermidis* induces CD8+ tissue-resident memory T (TRM) cells to produce IL-17 and Interferon gamma (IFN-γ), which in turn activate KC to produce AMPs. *S. epidermidis*-activated TRM cells upregulate the expression of KC mitogens, such as amphiregulin (AREG), fibroblast growth factor 2 (FGF2), and transforming growth factor beta 1 (TGFB1), to promote keratinocyte proliferation. In the later phase of wound healing, the engulfment of apoptotic NP by M is essential for M’s conversion into an anti-inflammatory phenotype and the production of several key factors for wound healing, including TGFB1, vascular endothelial growth factor (VEGF), and platelet-derived growth factor (PDGF).

**Figure 3 ijms-21-08790-f003:**
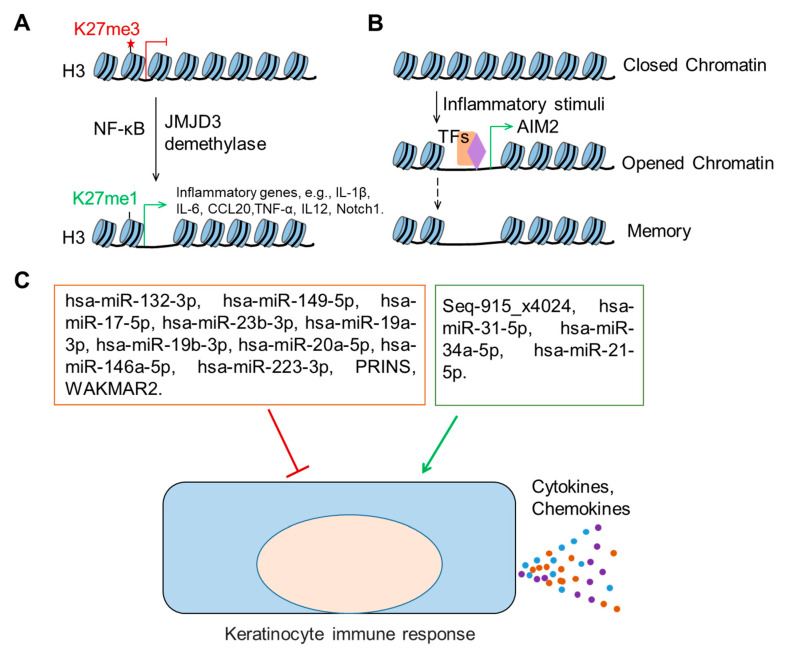
Epigenetic regulation of keratinocyte immune functions in wound healing. (**A**), The H3K27me3 demethylase Jumonji domain containing-3 (JMJD3), cooperating with the nuclear factor kappa-light-chain-enhancer of activated B cells (NF-κB), activates the expression of several inflammatory genes, including interleukin 1 beta (IL-1β), interleukin 6 (IL-6), chemokine (C-C motif) ligand 20 (CCL20), tumor necrosis factor (TNF-α), interleukin 12 (IL12), and Notch1 that is a crucial transcription factor in skin development and homeostasis. (**B**), Inflammatory stimuli trigger the chromatin opening in epithelial stem cells. After the resolution of inflammation, the chromatin accessibility is maintained for several critical inflammatory response genes, such as absent in melanoma 2 (AIM2), allowing the rapid transcription of AIM2 upon secondary stimulus. (**C**), Noncoding RNAs regulate keratinocyte immune response. The noncoding RNAs in the red frame negatively regulate cytokine or chemokine secretion in keratinocytes, while the ones in the green frame positively regulate them.

**Table 1 ijms-21-08790-t001:** Toll-like receptor (TLR)-mediated cytokine and chemokine expression in keratinocytes.

Keratinocyte Pattern-Recognition Receptors	Pathogen-Associated Molecular Patterns	Keratinocyte-Derived Inflammatory Mediators	References
TLR1/TLR2	Bacterial tri-acylated lipoproteins	Chemokine (C-C motif) ligand 20 (CCL20), CCL2, interleukin 18 (IL-8)	[14]
TLR2/6	*Staphylococcus aureus* peptidoglycan and lipoteichoic acid	IL-8, inducible nitric oxide synthase (iNOS)	[15]
TLR3	Double-strand RNA from viruses or damaged cells	Interferon Beta (IFN-β), IL-8, IL-18, Tumor necrosis factor (TNFα), IL-36γ, Chemokine (C-X-C motif) ligand 9 (CXCL9), CCL2, CCL20, CCL27	[9,16,17,18]
TLR4	Bacterial lipopolysaccharide	IL-1β, TNF-α, IL-8, CCL2, CCL20	[9]
TLR5	Bacterial flagellin	IL-8, TNF-α, CCL2, CCL20, CCL27	[9,16]
TLR9	Bacterial unmethylated CpG-containing DNA	TNF-α, IL-8, CXCL10, CCL2, CCL20	[9,12]

**Table 2 ijms-21-08790-t002:** MicroRNAs regulating keratinocyte immune functions in normal and chronic wounds.

MicroRNAs	Functions	Major Targets	References
hsa-miR-132-3p	Reduces keratinocyte-derived chemokines and cytokines while promoting keratinocyte proliferation and migration	HB-EGF ^1^	[127]
Seq-915_x4024	Inhibits keratinocyte-derived chemokines and cytokines while enhancing the proliferation of keratinocytes and their ability to promote fibroblast migration and growth	Sar1A, Smad2, TNF-α ^2^, and IL-8 ^3^	[137]
hsa-miR-149-5p	Downregulates keratinocyte inflammatory cytokine expression	IL-1α, IL-1β, and IL-6	[138]
hsa-miR-17-5p	Downregulates keratinocyte inflammatory cytokine expression	Not shown	[139]
hsa-miR-23b-3p	Inhibits keratinocyte-derived pro-inflammatory cytokines	ASK1 ^4^	[140]
hsa-miR-31-5p	Promotes keratinocyte proliferation and migration	EMP-1 ^5^	[141]
mmu-miR-31-5p	Promotes keratinocyte proliferation and migration	Rasa1 ^6^, Spred1 ^7^, Spred2, and Spry4 ^8^	[142]
dre-mir-223	Suppresses NF-κB ^9^ activation in basal epithelial cells to dampen neutrophil recruitment and inflammation	Cul1a ^10^, Cul1b, Traf6^11^, and Tab1 ^12^	[143]
hsa-miR-19a-3p and hsa-miR-19b-3p	Decreases TLR3 ^13^-mediated NF-κB activation in keratinocytes	SHCBP1 ^14^	[144]
hsa-miR-20a-5p	Decreases TLR3-mediated NF-κB activation in keratinocytes	SEMA7A ^15^	[144]
hsa-miR-146a-5p	Inhibits the NF-κB signaling pathway in keratinocytes	IRAK1 ^16^ and TRAF6	[145]
hsa-miR-34a-5p and hsa-miR-34c-5p	Promotes inflammatory chemokine and cytokine production by keratinocytes	LGR4 ^17^	[146]
hsa-miR-203a-3p	Suppresses skin re-epithelialization	RAN ^18^, RAPH1 ^19^, and IL-8	[147,148]
hsa-miR-21-5p	Promotes skin re-epithelialization	TIMP3 ^20^ and TIAM1 ^21^	[149]
hsa-miR-130a-3p	Suppresses skin re-epithelialization	LepR ^22^	[150]

^1.^ HB-EGF: Heparin Binding EGF Like Growth Factor; ^2.^ TNF-α: tumor necrosis factor; ^3.^ IL-8: interleukin 8; ^4.^ ASK1: Apoptosis signal-regulating kinase 1; ^5.^ EMP-1: Epithelial membrane protein 1; ^6.^ Rasa1: RAS P21 Protein Activator 1; ^7.^ Spred1: Sprouty Related EVH1 Domain Containing 1; ^8.^ Spry4: Sprouty RTK Signaling Antagonist 4; ^9.^ NF-κB: nuclear factor kappa-light-chain-enhancer of activated B cells; ^10.^ Cul1a: Cullin 1a; ^11.^ Traf6: TNF Receptor Associated Factor 6; ^12.^ Tab1: TGF-Beta Activated Kinase 1 (MAP3K7) Binding Protein 1; ^13.^ TLR3: Toll-like receptor 3; ^14.^ SHCBP1: SHC Binding And Spindle Associated 1; ^15.^ SEMA7A: Semaphorin 7A; ^16.^ IRAK1: Interleukin 1 Receptor Associated Kinase 1; ^17.^ LGR4: Leucine Rich Repeat Containing G Protein-Coupled Receptor 4; ^18.^ RAN: ras-related nuclear protein; ^19^^.^ RAPH1: Ras Association (RalGDS/AF-6) And Pleckstrin Homology Domains 1; ^20.^ TIMP3: TIMP Metallopeptidase Inhibitor 3; ^21.^ TIAM1: TIAM Rac1 Associated GEF 1; ^22.^ LepR: Leptin Receptor.

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
