# Peer review of "The Immune Functions of Keratinocytes in Skin Wound Healing"

_ijms, 2020, doi:10.3390/ijms21228790_

Round 1

Reviewer 1 Report

The manuscript by Piipponen and colleagues describes the communications between keratinocytes and immune cells in wound healing. They describe in detail the different cytokines, chemokines, antimicrobial peptides and extracellular vesicles involved in this process. Furthermore, they depict the epigenetic mechanisms known to regulate keratinocyte immune functions. Finally, they summarize the immune functions of keratinocytes in chronic nonhealing wounds.

The manuscript is well written, very clear and concise. It gives a very good overview of this complex topic which is relevant for many scientists. It will therefore help to understand and study keratinocytes’ immune functions in wound repair and establish innovative wound treatments.

However, I have some suggestions to further improve the manuscript:

  1. It would be helpful to show the crosstalk between keratinocytes and neutrophils and keratinocytes and macrophages described in section 2.1.3. also in Figure 2 in more detail, like it is shown for Langerhans cells, DECTs and CD8-T cells.
  2. The authors mention RIG-I as a PRR expressed by keratinocytes. They could describe the relevance of this receptor for the crosstalk between keratinocytes and immune cells in more detail.
  3. Cytosolic DNA which is associated with microbial infections is detected by the cGAS/STING pathway. The manuscript would benefit from a description of this pathway, as the activation of this pathway accelerates skin wound healing.
  4. There are some formatting errors with Greek letters (line 273, 277, 440, 441, 508)

Author Response

Point 1: It would be helpful to show the crosstalk between keratinocytes and neutrophils and keratinocytes and macrophages described in section 2.1.3. also in Figure 2 in more detail, like it is shown for Langerhans cells, DECTs and CD8-T cells.

Response 1: We have revised Figure 2 as suggested by the reviewer.

Point 2: The authors mention RIG-I as a PRR expressed by keratinocytes. They could describe the relevance of this receptor for the crosstalk between keratinocytes and immune cells in more detail.

Response 2: We include this part in the revised manuscript line 106-137.

Point 3: Cytosolic DNA which is associated with microbial infections is detected by the cGAS/STING pathway. The manuscript would benefit from a description of this pathway, as the activation of this pathway accelerates skin wound healing.

Response 3: We include this part in the revised manuscript line 106-137.

Point 4: There are some formatting errors with Greek letters (line 273, 277, 440, 441, 508)

Response 4: We have changed these in the revised manuscript.

Reviewer 2 Report

This review is well written and provides useful information with a commensurate degree of scientific rigour on a very hot topic that is immunomodulation and wound healing. Moreover, it offers a different approach, since rather than addressing the classic and well-known keratinocyte functions, it is centred around their involvement in immunomodulation of wounds, a topic much less talked about yet not less important. However, I put forwards some suggestions that I would like to be taken into consideration:

  • In Figure 2, I suggest labelling each different cell type in the ‘’tissue injury’’ and ‘’bacterial contact’’ for quicker understanding of the processes described in the figure.
  • In line 266, ‘’ Several studies have shown the pro-healing effects of exosomes in animal models with impaired healing capacity.’’ This statement is supported by a single reference from a review on this topic, please add the original articles to provide a better access to said studies.
  • Please remove any reference to ‘’your unpublished study’’. Line 284. Such practice is not allowed by scientific community.
  • The inclusion of another figure in section 4, ‘’Epigenetic regulation of keratinocyte immune functions in wound healing’’, would not only help with the visual appearance of this manuscript but also give the reader a quick grasp of the content explained in this section.
  • In line 440 and 441, the Greek symbols have been replaced by spirals.

Author Response

Point 1: In Figure 2, I suggest labelling each different cell type in the ‘’tissue injury’’ and ‘’bacterial contact’’ for quicker understanding of the processes described in the figure.

Response 1: We have revised Figure 2 as suggested by the reviewer.

Point 2: In line 266, ‘’ Several studies have shown the pro-healing effects of exosomes in animal models with impaired healing capacity.’’ This statement is supported by a single reference from a review on this topic, please add the original articles to provide a better access to said studies.

Response 2: We have added several original articles as references to support this description. Please see line 304 in the revised manuscript.

Point 3: Please remove any reference to ‘’your unpublished study’’. Line 284. Such practice is not allowed by scientific community.

Response 3: We have revised this part as suggested by the reviewer.

Point 4: The inclusion of another figure in section 4, ‘’Epigenetic regulation of keratinocyte immune functions in wound healing’’, would not only help with the visual appearance of this manuscript but also give the reader a quick grasp of the content explained in this section.

Response 4: As suggested by the Reviewer, we have added Figure 3 to summarize the section 4.

Point 5: In line 440 and 441, the Greek symbols have been replaced by spirals.

Response 5: We have changed these in the revised manuscript.